# Evaluation of Polymeric Particles for Modular Tissue Cultures in Developmental Engineering

**DOI:** 10.3390/ijms24065234

**Published:** 2023-03-09

**Authors:** Yu Xiang, Jiongyi Yan, Xujin Bao, Andrew Gleadall, Paul Roach, Tao Sun

**Affiliations:** 1Department of Materials, Loughborough University, Epinal Way, Loughborough LE11 3TU, UK; 2Wolfson School of Mechanical, Electrical and Manufacturing Engineering, Loughborough University, Epinal Way, Loughborough LE11 3TU, UK; 3Department of Chemistry, Loughborough University, Epinal Way, Loughborough LE11 3TU, UK; 4Department of Chemical Engineering, Loughborough University, Epinal Way, Loughborough LE11 3TU, UK

**Keywords:** developmental engineering, modular tissue culture, polymeric particle, modular scaffold, cell colonization

## Abstract

Developmental engineering (DE) aims to culture mammalian cells on corresponding modular scaffolds (scale: micron to millimeter), then assemble these into functional tissues imitating natural developmental biology processes. This research intended to investigate the influences of polymeric particles on modular tissue cultures. When poly(methyl methacrylate) (PMMA), poly(lactic acid) (PLA) and polystyrene (PS) particles (diameter: 5–100 µm) were fabricated and submerged in culture medium in tissue culture plastics (TCPs) for modular tissue cultures, the majority of adjacent PMMA, some PLA but no PS particles aggregated. Human dermal fibroblasts (HDFs) could be directly seeded onto large (diameter: 30–100 µm) PMMA particles, but not small (diameter: 5–20 µm) PMMA, nor all the PLA and PS particles. During tissue cultures, HDFs migrated from the TCPs surfaces onto all the particles, while the clustered PMMA or PLA particles were colonized by HDFs into modular tissues with varying sizes. Further comparisons revealed that HDFs utilized the same cell bridging and stacking strategies to colonize single or clustered polymeric particles, and the finely controlled open pores, corners and gaps on 3D-printed PLA discs. These observed cell–scaffold interactions, which were then used to evaluate the adaptation of microcarrier-based cell expansion technologies for modular tissue manufacturing in DE.

## 1. Introduction

Developmental engineering (DE) aims to manufacture multiple tissue building block materials or modular tissues via culturing different types of mammalian cells on corresponding modular scaffolds (scale: micron to millimeter), then assemble these modular tissues layer-by-layer into large and more functional tissues, mimicking the natural developmental biology processes [1,2,3,4]. To produce the required multiple modular tissues, three-dimensional (3D) modular scaffolds with suitable structures need to be fabricated for different cell types [5]. For example, previous research in tissue engineering and 3D tissue cultures demonstrated that tubular or mesh polymeric scaffolds are suitable for blood vessel cells [6,7,8,9], spherical and/or porous scaffolds for skin cells [10,11], while nanofibers are suitable for nerve cells and human glioma cells [12,13,14]. In addition, the scaffolds with other forms such as membranes [15], sponges [16] and woven fabrics [17] are also fabricated according to the cell types and the targeted tissues. For the gradual layer-by-layer reconstruction of large and more functional 3D tissues with the equivalent sizes (scale: millimeter to meter) and structures of the target tissues, it is also necessary to manufacture sufficient various modular tissues via culturing different types of cells on the suitable modular scaffolds at large scale, which is comparable to the current microcarrier-based cell expansion systems [18,19,20,21]. 

Microcarriers can be solid spherical matrices ranging between 30–860 µm in diameter [11,22,23,24], which are usually synthesized with either synthetic materials such as polylactic-co-glycolic acid [25] and polycaprolactone [11], or natural materials such as gelatin [10,26]. As adherent mammalian cells can be seeded and further cultured on the curved surfaces provided by enormous microcarriers suspended in bioreactors, the cell expansion capacity is dramatically enhanced [27,28,29,30]. Apart from solid spherical particles, octopus-shaped [31], asymmetric dumbbell-shaped [32], and porous microcarriers [33,34] are also manufactured for large-scale cell cultures. The advantages of these microcarrier-based expansion systems include: (i) ease of scale up [29,35,36,37], (ii) ability to precisely control cell growth conditions within large-scale bioreactors [38,39], (iii) reduction in bioreactor volumes and the floor spaces required for given-sized manufacturing operations [40], (iv) reduction in technician labor [37,39], and (v) more natural culture environments for cell proliferation and differentiation [38,41]. All these benefits have enabled the microcarrier-based culture technology to be generally employed for industrial production of mammalian cells [37,42], protein-based therapeutics [29], and for research purposes [35]. Furthermore, it potentially provides a straightforward scale-up strategy for modular tissue cultures in DE. Our aim here was to investigate the influences of material type and size characteristics of polymeric scaffolds on modular tissue cultures at small scale, then evaluate their potential impacts on the adaption of current microcarrier-based cell culture technologies for large-scale modular tissue manufactures. 

For the ease of systematic comparisons, three distinct polymers, i.e., amorphous hydrophilic poly(methyl methacrylate) (PMMA), hydrophobic poly(lactic acid) (PLA) with certain crystallinity, and amorphous hydrophobic polystyrene (PS), were used to fabricate spherical modular scaffolds with different sizes (diameter: 5–100 µm). Human dermal fibroblasts (HDFs) were selected as the exemplar cell for small-scale modular tissue cultures in tissue culture plastics (TCPs). The influences of scaffold materials and particle sizes on the initial cell seedings and the following modular tissue cultures were investigated. The self-assembly of the polymeric modular scaffolds with or without cultured cells, the strategies used by HDFs to colonize the single particles with different sizes, the aggregated spherical scaffolds and the finely controlled regular open structures on 3D-printed thin PLA discs were intentionally compared. The potential impacts of these mechanistic insights on the design of large-scale modular tissue manufacture processes were then evaluated.

## 2. Results

### 2.1. Influences of Scaffold Material and Particle Size on Modular Tissue Cultures

In the preliminary modular tissue cultures, PMMA, PLA or PS particles with different diameters (5, 10, 20, 30 or 100 µm) were prepared to fully cover the flat surfaces of each well in 24-well tissue culture plates and then submerged in 1 mL DMEM, respectively. Aliquots of 1 mL HDFs (10^3^ cells/mL) were then seeded into each well with different polymeric modular scaffolds. Phase contrast microscopic analysis indicated that the HDFs were seeded onto all these particles, and it was not possible to inspect the impacts of scaffold materials and sizes on the initial cell seedings. Consequently, in the following modular tissue cultures, a defined amount (5 mg) of PMMA, PLA or PS particles with different diameters was prepared and situated in each well of 24-well plates. It was observed that most of the neighboring PMMA particles and a few adjacent PLA particles agglomerated in the DMEM medium, while no PS particles aggregated. Aliquots of 1 mL HDFs (10^3^ cells/mL) were then seeded into each well with single and/or clustered polymeric modular scaffolds. Both phase contrast and fluorescent microscopic analysis showed that the majority of the HDFs were initially seeded onto the flat surfaces of the cell culture plates; some cells were also directly seeded onto the large PMMA particles (diameter: 30–100 µm), while almost no cells were seeded onto the smaller PMMA particles (diameter: 5–20 µm), nor were all the investigated PLA or PS particles with different diameters (5–100 µm). After being further cultured for 2–6 days, the HDFs on the flat TCPs surfaces proliferated and migrated randomly; some cells also migrated onto the neighboring spherical particles (Figure 1I). Consequently, the single (Figure 1(IA,J)) and clustered PMMA particles (Figure 1(IB,C,K,L)), as well as the single (Figure 1(ID,M)) and aggregated PLA particles (Figure 1(IE,F,N,O)) were all covered and further wrapped up by HDFs to form modular tissues with varying sizes and structures. In contrast, the non-clustered single PS particles were colonized by HDFs as shown in Figure 1(IG–I,P–R).

Comparison of DAPI-stained nuclei (blue) and phalloidin-stained cytoskeleton (green) in the fluorescent micrographs (Figure 1(IJ–R)), and further detailed scanning electron microscopic (SEM) analysis indicated that HDFs migrated from flat TCPs surfaces onto relatively large polymeric particles (diameter: 30–100 µm) via cell bridges formed by either single cell (Figure 1(IIA,H,M,N)) or multiple cells (Figure 1(IIB,I,P)), which were then expanded into cell membranes as more cells joined in and stacked together (Figure 1(IIC,J,Q,R)). In contrast, the smaller spherical modular scaffolds (diameter: 5–20 µm) were totally covered by single spreading HDFs as shown in Figure 1(IIG). To migrate across the aggregated PMMA or PLA particles, the same cell bridging (Figure 1(IID,K)) and cell stacking (Figure 1(IIE,F,L)) strategies were applied. Depending on the number and size of the clustered PMMA particles, 3D tissues with varying sizes and structures were created by the cells colonized in the open spaces among these particles (Figure 1(IID,F)). In comparison, as fewer number of PLA particles agglomerated, relatively smaller 3D tissues were produced by the colonized cells (Figure 1(IIK,L)). Sirius red staining of the HDFs cultured on different polymeric particles indicated that all the cells colonized on the spherical modular scaffolds and the gaps between these particles expressed collagens (Figure 2I), suggesting the mechanical interaction between the cells and the polymeric particles. 

After being cultured for more than 6–10 days, it was observed that some of the singular cells and cell membranes initially attached on the flat surfaces of the tissue culture plates were peeled off by most of the neighboring large or agglomerated PMMA particles (Figure 2(IIA–C)). In contrast, the single or clustered PLA particles (Figure 2(IID–F)) and all the PS particles (Figure 2(IIG–I)) demonstrated no obvious influences on the neighboring two-dimensional (2D) cell cultures on the flat TCPs surfaces.

### 2.2. Influences of Particle Material and Number on Neighboring 2D Cell Cultures

To quantify the impacts of particle materials and numbers on the neighboring 2D cell cultures on flat TCPs surfaces and the formation 3D modular tissues, different amounts of PMMA, PLA or PS particles (1, 5, 10 or 20 mg) with the defined diameter (100 µm) were prepared and added into each of T25 flasks containing 5 mL DMEM media, while T25 flasks containing no polymeric particles were used as the controls. Aliquots of 1 mL HDFs (10^5^ cells/mL) were seeded and cultured in each of these flasks for 14–30 days. The cell populations on the TCPs surfaces were estimated via PCM on a daily basis. In the no particle controls, complete cell confluency was reached after being cultured for 11–13 days (Figure 3I). When only 1 mg of PMMA particles were utilized for small-scale modular tissue cultures in each T25 flask, particle aggregation was not very obvious; the cells on TCPs surfaces were only influenced by singular PMMA particles and reached complete confluency after cultured for 25–30 days (Figure 3I). When more than 5 mg PMMA particles were utilized, more adjacent particles aggregated; some of the cells on TCPs surfaces were affected by both singular and aggregated particles. After cultured for 14–15 days, the cell population on flask surfaces started to increase; meanwhile, some of the clustered PMMA particles further aggregated due to HDF colonization on these particles, which peeled the cell membranes off the flat TCPs surfaces and caused sharp reductions in the neighboring 2D cell cultures at day 22–23. As culture continued, HDFs re-colonized the flat flask surfaces and the populations of 2D cell cultures increased again. In comparison, the impacts of different amounts of PLA or PS particles on the neighboring 2D cell cultures on TCPs surfaces were not detectable, and complete cell confluency in each of these flasks was reached after being cultured for 10–12 days (Figure 3II,III).

### 2.3. Influences of Particle Material and Size on Neighboring 2D Cell Cultures

To quantify the impacts of particle materials and sizes on the neighboring 2D cell cultures on flat TCPs surfaces, aliquots of 5 mg PMMA, PLA or PS particles with different diameters (5, 10 and 100 µm) were prepared and situated in each of T25 flasks containing 5 mL DMEM media, respectively, while T25 flasks containing no polymeric particles were used as the controls. Aliquots of 1 mL HDFs (10^5^ cells/mL) were then seeded and cultured in each flask for 14 days. PCM analysis on daily basis indicated that it took 11–13 days for the cells in the control groups to reach complete confluency, as illustrated in Figure 3I. PCM analysis of the cells in the flasks with polymeric particles demonstrated that the cell adherences and proliferations on the flat surfaces were very similar when smaller PMMA, PLA or PS particles (diameter: 5 and 10 µm) were used for modular tissue cultures, and the total cell confluency in each of these flasks was reached after being cultured for 6–7 days (Figure 3IV–VI). However, when large particles (diameter: 100 µm) were used, the polymeric materials demonstrated significant impacts on the neighboring 2D cell cultures as very different cell adhesions and growths were observed in the T25 flasks containing PMMA, PLA or PS particles. For example, very low cell density was detected on the surfaces of T25 flasks with PMMA particles (diameter: 100 µm) (Figure 3IV) due to the influences of the singular and aggregated PMMA particles. In contrast, the cell adherences and proliferations on the flask surfaces were not obviously affected by large PLA and PS particles (diameter: 100 µm). However, it took a relatively longer time (9–11 days) for the 2D cell cultures on the surfaces of these flasks to reach complete confluency (Figure 3V,VI), in comparison with the cells on the flat surfaces of the flasks containing smaller PLA or PS particles (diameter: 5 and 10 µm).

In order to investigate the effects of the surface properties of these polymeric particles, the contact angles of PMMA, PLA and PS with different concentrations (0.5, 5 and 62.5 mg/mL) were measured and compared. The glass slides without any casted polymeric films or 0 concentration of polymer were used as the controls. As shown in Figure 4I, the contact angles of the glass slide control, the PMMA, PLA and PS values within the investigated concentration range were 38–42°, 67–74°, 83–100° and 102–108°, respectively, indicating the relatively high hydrophilicity of PMMA, low hydrophilicity of PLA and high hydrophobicity of PS. To further evaluate the impacts of their surface properties, the zeta potentials of PMMA, PLA and PS particles with different diameters (5, 10, 20, 30 and 100 µm) were then measured in deionized water. It was demonstrated that the polymeric materials had detectable influence on the exhibited surface charges, as shown in Figure 4II. For the PMMA, PLA and PS particles with a diameter of 100 µm, the zeta potentials were −35.50, −0.52 and −0.03 mV, respectively, which were changed to −5.11, −1.61 and −0.54 mV, respectively when the particle diameter was reduced to 5 µm. Moreover, there was a clear inverse dependence of the zeta potential on the PMMA particle sizes, as it gradually increased from −35.50 to −5.11 mV when the diameter was decreased from 100 to 5 µm. In contrast, the zeta potential of PLA and PS particles changed linearly with the particle diameters; however, this dependency was inconsistent and not statistically significant.

### 2.4. Influences of PLA Particle Surface Modifications on Modular Tissue Cultures

Due to its hydrophilicity being detected to be lower than that of PMMA but higher than that of PS, PLA was selected for further surface modifications and then compared with the non-treated particles. PLA particles (diameters: 50–100 µm) were produced and divided into 6 groups for separate surface modifications. Apart from the non-treatment controls (PLA), other PLA particles were treated via (i) NaOH: hydrolysis in 0.5 M sodium hydroxide (NaOH) at room temperature (RT) for 24 h, (ii) PLL: immersion in poly-L-lysine (PLL) (10 µg/mL) at 4 °C for 24 h; (iii) FBS: immersion in fetal bovine serum (FBS) at 4 °C for 24 h; (iv) NL: immersion in PLL after NaOH hydrolysis; (v) NF: immersion in FBS after NaOH hydrolysis. Aliquots of 5 mg of the treated or non-treated PLA particles were prepared and placed in each well of 24-well culture plates containing 1 mL DMEM media respectively. It was observed that more of the NF- and NL-treated adjacent PLA particles aggregated when submerged in DMEM media in comparison with the other treated and non-treated spherical modular PLA scaffolds. Aliquots of 1 mL HDFs (10^4^ cells/mL) were then seeded into each well and incubated for different time periods. MTT assay of the cells cultured at 6, 12 and 24 h indicated that significantly higher cell viability was observed on NL-treated PLA particles in comparison with other treated and non-treated particles (Figure 5I). Interestingly, the cell viabilities on the NF-treated PLA particles at 6 and 12 h were relatively low, while higher cell viability was detected after cultured for 24 h. PCM analysis of the modular tissues cultured for 14 days demonstrated (Figure 5II) that HDFs infiltrated onto the clustered NL- and NF-treated PLA particles (Figure 5(IIE,F)), while the majority of the PLA particles treated using other methods were not observed to aggregate before or after the modular tissue cultures (Figure 5(IIA–D)).

### 2.5. Cell Colonization within Finely Controlled Open Structures

As HDFs were observed to migrate onto the single and clustered particles via cell bridging and/or cell stacking, the infiltrations of HDFs onto other open structures were further investigated. For comparison purposes, circular PLA discs (diameter: 6 mm: thickness: 500 µm) with finely controlled open pores (diameter: 400, 500, 640 and 1100 µm), corners (angle: 40, 50, 100 and 120°), and gaps (distance: 130, 170, 320, 510 and 970 µm) were prepared in each well of 24-well tissue culture plates. After seeded with HDFs on each top surface, the PLA discs were submerged in DMEM media for further tissue cultures. PCM analysis demonstrated that HDFs utilized the same cell bridging and/or cell stacking as observed in the modular tissue cultures to infiltrate and colonize the corners (Figure 6(IA–F)), gaps (Figure 6(IG–I)) and circular open pores (Figure 6(IJ–L)) after being cultured for 7–20 days. Further SEM examination illustrated that HDFs initially colonized the top PLA surfaces, then migrated from flat surfaces into the regular open structures (Figure 6(IIA–C)), colonized the corners (Figure 6(IID–F)), gaps (Figure 6(IIG–I)) and circular open pores (Figure 6(IIJ–L)) via cell bridges and/or large cell membranes formed via stacked cells.

## 3. Discussion

The impacts of spherical PMMA, PLA and PS scaffolds with varying sizes on small-scale modular tissue cultures were evaluated in this study. It was discovered that both particle materials and sizes could affect the initial seeding of the exemplar HDFs on these spherical modular scaffolds. The impacts of the supporting matrix materials or surface properties on the adhesion, spreading, proliferation and even survival of adherent mammalian cells have been generally investigated [43,44,45]. Various mechanisms have also been proposed for cell adherences on the supporting matrix. It can be facilitated by the electrostatic attraction induced by the positively or negatively charged mammalian cells and the supporting matrix [46]. For example, the negatively charged osteoblast cells [47], 3T3 fibroblast cells [46], and MG-63 cells [48] could attach to the positively charged scaffolds, while the negatively charged surfaces were beneficial to the adhesions of the positively charged human fetal osteoblasts [49] and the differentiation of chondrocytes [43]. Cell adhesions are also prompted by the proteins, peptides or other molecules coated on the surfaces of the supporting matrix [44,45], while the coating processes are dependent on the surface wettability of the biomaterials [44,45]. It was discovereddiscovered that most of the adherent cells such as 3T3 fibroblasts [50] and human bone marrow stromal cells [51] prefer hydrophilic surfaces, while maximum HDFs adhesion was achieved on the surfaces with the contact angles of 60–80° [43]. This was also confirmed by our cell seeding experiments using different polymeric particles and specific contact angle and zeta potential measurements. 

Apart from scaffold materials or surface properties, cell adhesions are also dependent on other factors including scaffold sizes [43,52,53,54] and specific experimental protocols. For example, the electro-spun fibers of >10 μm in diameter were previously observed to be suitable for HDFs [55]. This was because the single fibers with varying diameters were suspended in culture media, and the cells were only allowed to migrate from the cell seeding chambers onto the suspended fibers, which was not affected by the underneath flat tissue culture surfaces of the TCPs. In order to examine the impacts of particle materials and sizes on initial cell seedings and the following modular tissue cultures, defined amount of PMMA, PLA and PS particles were deliberately utilized to ensure the flat surfaces of TCPs (i.e., the tissue culture plates or flasks) were not fully covered. The existence of the extra 2D flat TCPs surfaces made it possible to compare and contrast the effects of different polymeric particles on the initial cell seedings in small-scale modular tissue cultures. Consequently, apart from the large PMMA particles with suitable wettability (contact angle: 67–74°) for HDFs attachment [43], the majority of the HDFs were seeded onto TCPs surfaces. However, the HDFs were observed to migrate from the TCPs surfaces onto all the tested PMMA, PLA and PS particles. The migration and proliferation of HDFs on these polymeric particles might be facilitated by the extra cellular matrix proteins such as collagen [50,56] produced by the HDFs migrated from the TCPs onto these particles. The influences of surface properties and/or the extra cellular matrix proteins were also confirmed by our comparisons of the PLA particles treated via different surface-modification methods.

The observed effects of modular scaffold materials and sizes on the initial cell adhesions are essential to inform the adaptation of current microcarrier-based cell culture technologies for large-scale modular tissue cultures. In the microcarrier-based cell expansion systems, adherent mammalian cells are usually seeded on the microcarriers with suitable surface properties either continuously suspended in the media or precipitated statically in the bioreactors [27,57,58,59]. The selection of either the dynamic or the static cell seeding methods is dependent on several issues including the size of the microcarriers. It was revealed that the dynamic method is generally used for larger microcarriers (>100 µm) [28,57,58,60], while the static method is frequently selected for smaller microcarriers (56–100 µm) [61,62,63]. According to the size and surface properties of the modular scaffolds, these dynamic or static cell seeding methods designed for microcarrier-based cell expansion systems can be adapted for large-scale modular tissue cultures in DE. Our cell seeding and culture experiments on three distinct polymeric materials suggested that the cell seeding efficiency was dependent on particle size and seeding methods. The static cell seeding method can be further evaluated for PLA or PS particles with different sizes and smaller PMMA (<30 µm), while the dynamic cell seeding method might be suitable for the large PMMA particles (>30 µm). 

One of the primary concerns in the microcarrier-based culture systems for large-scale cell expansions is microcarrier aggregation [64,65,66,67], which is largely driven by cell bridging and/or cell migration from donor to acceptor beads [18,68,69]. Our research suggested that the surface properties also contributed to the modular scaffold aggregations. For example, in comparison with the hydrophobic PS particles, some of the less hydrophilic PLA particles and most of the hydrophilic PMMA particles aggregated in DMEM, which were further colonized and wrapped by HDFs via cell bridging and/or cell stacking strategies in the following modular tissue cultures. It was further demonstrated that the aggregation of some of the PLA particles could be enhanced via suitable surface modifications. The observed influences of different surface properties on particle aggregation and subsequent cell infiltration and colonization might also be due to the denaturation or conformational changes of the proteins adsorbed from the cell culture medium (e.g., the supplement fetal bovine serum) onto the hydrophilic or hydrophobic particles [70].

In contrast to microcarrier-based culture systems, the self-assembly or aggregation of the modular tissues can be further exploited to manufacture modular tissues with different sizes and structures, which will facilitate and simplify the following layer-by-layer assembly of large and more functional tissues in DE. Apart from colonizing the clustered polymeric particles, our comparison research demonstrated that the same cell bridging and cell stacking strategies were also applied by HDFs to migrate from the flat TCPs surfaces onto the polymeric particles, and to infiltrate and colonize the finely controlled open structures on 3D-printed PLA discs. Therefore, the same cell bridging and stacking strategies should be employed by HDFs for the infiltration and colonization of modular scaffolds with different shapes and sizes, which can be used for further gradual assembly of large and more functional tissues in DE.

## 4. Materials and Methods

### 4.1. Fabrication of Polymeric Particles

Poly(methyl methacrylate) (PMMA, average molecular weight (Mw): 120,000 g/mol), polystyrene (PS, Mw: 350,000 g/mol), poly(lactic acid) (PLA, PLA 4042D, Mw: 120,000 g/mol) and poly(vinyl alcohol) (PVA, Mw: 13,000–23,000 g/mol, 87–89% hydrolyzed) were purchased from Sigma-Aldrich (UK) and used as received. Chloroform and dichloromethane (DCM) were purchased from Fisher Scientific Inc. (Loughborough, UK). Polymeric particles with various diameter (5, 10, 30 and 100 µm) were prepared via emulsification and solvent diffusion/evaporation method. As shown in Table 1, the organic solutions (5 or 62.5 mg/mL) were prepared by dissolving PMMA, PLA or PS in DCM or chloroform; the aqueous phase was prepared by dissolving PVA surfactant in distilled water to defined concentrations (0.1, 1 or 5% (*w*/*v*)). The specific organic solution was then injected into the aqueous phase (with 8:150 or 1:10 ratios) under magnetic stirring (200 or 1500 RPM (revolutions per minute)), which was followed by solvent removal via evaporation or distillation at room temperature (RT) or 65 ˚C for 2.5 or 1.5 h. The precipitated polymeric particles were collected via centrifugation (300 RCF (relative centrifugal force) for 15 min at RT) and washed thoroughly using deionized water (×3) to remove the surfactant.

### 4.2. 3D Printing of PLA Discs with Defined Open Structures

Circular PLA discs (diameter: 6 mm; thickness: 500 µm) with circular open pores (diameter: 400, 500, 640 and 1100 µm), corners (angle: 40, 50, 100 and 120°), and gaps (distance: 130, 170, 320, 510 and 970 µm) were printed using a Raise3D Pro 3 printer (Raise3D^®^, Irvine, CA, USA) with a hardened steel nozzle (diameter: 0.4 mm). An open-source software (FullControl GCode Designer [71]) was used to create toolpaths of the target PLA discs, and directly generate manufacturing code (GCode), using the parameters shown in Table 2.

### 4.3. 2D Cell Culture

The medium used for the culture of Neonatal foreskin human dermal fibroblasts (HDFs, Intercytex, Manchester, UK) was Dulbecco’s modified Eagle’s medium (DMEM, Lonza, Walkersville, MD, USA) containing 4.5 g/L glucose supplemented with 2 mM L-glutamine (Gibco, Paisley, UK), 100 IU/mL penicillin and 100 μg/mL streptomycin (Gibco, Grand Island, New York, NY, USA), and 10% (*v*/*v*) fetal bovine serum (FBS, Gibco, Paisley, UK). HDFs were cultivated in tissue culture plates or T-flasks at 37 °C in 5% CO_2_ humidified atmosphere; the media in the plates or flasks were changed every three days. When 80–100% confluence was reached, the cells cultured in the T-flasks were detached using trypsin/EDTA (0.25% (*w*/*v*) solution, Gibco, Paisley, UK), collected via centrifugation at 300 RCF for 5 min, resuspended in fresh medium with certain density, then either used for specific experiments or continually passaged using flasks.

### 4.4. Small-Scale Modular Tissue Cultures on Spherical Modular Scaffolds

A defined amount of PMMA, PS or PLA particles with varying diameters was added into each well of 24-well culture plates or T-25 flasks, sterilized with 70% ethanol overnight, thoroughly rinsed with phosphate-buffered saline (PBS, Lonza, Walkersville, MD, USA) (×3), and submerged in DMEM media as the spherical modular scaffolds. HDFs with defined volume and cell density (cells/mL) were seeded into each well of 24-well tissue culture plate or T-25 flaks and cultured at 37 °C in 5% CO_2_ humidified atmosphere for different time periods. The media in the plates or flasks were changed every three days. During and after the modular tissue cultures, the cells were analyzed via phase contrast microscopy (PCM), fluorescence microscopy or scanning electron microscopy (SEM), respectively.

### 4.5. 3D Tissue Cultures on PLA Discs with Defined Open Structures

The PLA discs were sterilized with 70% ethanol overnight, washed thoroughly with PBS (×3) and dried at RT. Aliquot of 50 µL of HDFs (10^6^ cells/mL) was seeded on the top surface of each PLA disc and incubated statically at 37 °C in 5% CO_2_ humidified atmosphere for 1 h for firm cell attachment. Each of these PLA discs was then submerged in 1 mL of media in 24-well culture plate to culture the cells at 37 °C in 5% CO_2_ humidified atmosphere for 40–60 days; the media in the 24-well plates were changed every three days. During and after cell culture, the cells were analyzed via PCM or SEM, respectively.

### 4.6. MTT Viability Assay

The cells seeded onto the surface-modified PLA particles were compared using MTT (3-(4,5-dimethylthiazol-2-yl)-2,5-diphenyl tetrazolium bromide) viability assay as described previously [72,73]. Briefly, the cells cultured on the surface-modified PLA particles in each well of 24-well culture plates were rinsed with PBS (×3), then incubated in 200 µL MTT solution (5 mg/mL, Life Technologies Corporation, Eugnene, OR, USA) at 37 °C for 3 h. After careful removal of the MTT solution, 200 μL dimethyl sulfoxide (DMSO, Fisher Scientific Inc., Loughborough, UK) solution was added into each well to dissolve the formazan crystals formed in the viable cells. Aliquots of 50 μL DMSO solution from each well were then transferred into 96-well plate for optical density measurement at 570 nm (OD_570_) using a microplate spectrophotometer (BioTek instruments Ltd., Winooski, VT, USA).

### 4.7. Phase Contrast Microscopy

The cells cultured on tissue culture plastics, polymeric particles or PLA discs were monitored and analyzed noninvasively using an inverted phase contrast microscope (Eclipse TS100-F, Nikon Corporation, Shanghai, China) during cell or tissue cultures. The cell populations were estimated via analysis of the captured PCM images using ImageJ.

### 4.8. Fluorescence Microscopy

Fluorescence microscopy was performed after small-scale modular tissue cultures by staining the cultured HDFs with DAPI (400 nM, Life Technologies Corporation, Eugnene, OR, USA) and phalloidin (5 U/mL in PBS with BSA (bovine serum albumin), Life Technologies Corporation, Eugnene, OR, USA). In brief, after removal of the culture medium, the cultured cells were washed gently with PBS (×3), fixed in intracellular (IC) fixation buffer (Fisher Scientific, Loughborough, UK) for 10 min, rinsed thoroughly with PBS (×3), and then labelled with DAPI for 10 min. After being rinsed thoroughly with PBS (×3), the cells were then stained with phalloidin for 1 h. After further rinsing with PBS (×3), fluorescent images of the cells were captured at λ_ex_ = 360 nm, λ_em_ = 460 nm (for DAPI/nuclei visualization), at λ_ex_ = 495 nm, λ_em_ = 518 nm (for phalloidin/cytoskeleton visualization) using a fluorescent microscope (Nikon Eclipse Ti, Scientifica, UK).

### 4.9. Scanning Electron Microscopy

The cells cultivated on TCPs, polymeric particles or PLA discs were gently washed with PBS (×3), fixed in IC fixation buffer for 10 min, thoroughly washed with distilled water, left to dry at RT, and then coated with Gold/Palladium (Au/Pd) for 90 s using a splutter coater (Quorum Q150R S, Laughton, UK). In situ analysis of the cells was then conducted via SEM (JSM-7100F FE-SEM (field emission scanning electron microscope), Singapore) using In-lens mode with 5.0 kV accelerating beam.

### 4.10. Collagen Analysis

The cells cultivated on polymeric particles were gently rinsed with PBS (×3), and then fixed in 70% ethanol for 15 min. After being thoroughly washed with PBS (×3) and distilled water (×3), the samples were submerged in Picro-Sirius Red Solution (1.2% Picric Acid, 0.1% Direct Red 80, Abcam, UK) for 1 h, washed with 0.5 M acetic acid (×2), and then dehydrated in 100% ethanol (×3). The stained collagen (red) was analyzed via PCM. 

### 4.11. Contact Angle Measurement

Polymeric films were prepared by manually casting organic solutions on glass slides, which was followed by complete solvent evaporation at RT for 48 h. The contact angles of the polymeric films were measured via a data physics OCA (optical contact angle, Germany) system at 25 °C and 20% relative humidity. Briefly, 2 μL of deionized water was deposited onto the surface of each polymeric film. Within 3 s of each deposition, the static contact angle images were captured using a charge coupled device (CCD) camera. The average contact angles were then calculated via Dataphysics SCA 20 OCA control software (Germany) based on the captured images of the liquid droplets deposited on 5–8 independent sites of each polymeric film. The glass slides without any casted polymeric films or 0 concentration of polymers were used as the controls.

### 4.12. Zeta Potential Measurement

The effective surface charges of the polymeric particles with varying sizes were characterized by zeta potential using dynamic light scattering and zeta potentiometry (Malvern Zetasizer Ultra, UK) [74,75].

### 4.13. Statistical Analysis

All the experimental results were shown as mean ± standard deviation (SD) from at least three independent replicate experiments (*n* ≥ 3). One-way analysis of variance (ANOVA) was used for statistical significance analysis (* *p* < 0.05, ** *p* < 0.01, *** *p* < 0.001).

## 5. Conclusions

In this research, PMMA, PLA and PS particles with varying diameters (5–100 µm) were fabricated as the spherical modular scaffolds for modular tissue cultures. When initially submerged in DMEM media, the majority of the adjacent PMMA particles and some neighboring PLA particles were observed to aggregate into clusters. The particle materials and size also impacted the seeding of the exemplar HDFs on these spherical modular scaffolds situated in TCPs. Subsequent modular tissue cultures revealed that the HDFs utilized cell bridging and/or stacking strategies to migrate from the flat TCPs surfaces onto all the single and aggregated particles, and infiltrated and colonized the clustered PMMA and PLA particles into modular tissues with different sizes and structures. It was also demonstrated that the same cell bridging and stacking strategies were used by HDFs to infiltrate and colonize finely controlled open pores, corners and gaps on 3D-printed PLA discs. These cell–scaffold interactions observed from the small-scale modular tissue cultures were also used to evaluate the potential to adapt the microcarrier-based culture technologies for the manufacture of modular tissues in DE.

## Figures and Tables

**Figure 1 ijms-24-05234-f001:**
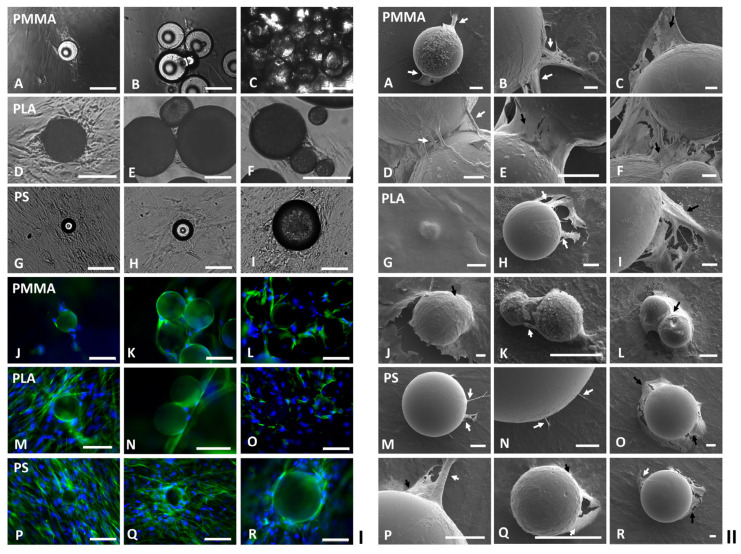
(**I**,**II**) Micrographs of human dermal fibroblasts (HDFs) cultured on spherical modular scaffolds. Aliquots of 1 mL HDFs (10^3^ cells/mL) were seeded in each well of 24-well tissue culture plate containing 5 mg poly(methyl methacrylate) (PMMA), poly(lactic acid) (PLA) or polystyrene (PS) particles (diameter: 5, 10, 20, 30 or 100 µm) submerged in 1 mL DMEM and cultured for 14–30 days. The phase contrast micrographs of HDFs (**I**) cultured on (**A**–**C**) PMMA, (**D**–**F**) PLA and (**G**–**I**) PS particles; the fluorescent micrographs of HDFs (**I**) cultured on (**J**–**L**) PMMA, (**M**–**O**) PLA and (**P**–**R**) PS particles were captured. Blue: nuclei stained by DAPI, Green: cytoskeleton stained by phalloidin. (scale bar = 100 µm). Scanning electron micrographs of HDFs (**II**) cultured on (**A**–**F**) PMMA, (**G**–**L**) PLA and (**M**–**R**) PS particles were captured. Cell colonization strategies via cell bridging and cell stacking are highlighted via White and Black arrows respectively. (scale bar = 10 µm except G = 1 µm).

**Figure 2 ijms-24-05234-f002:**
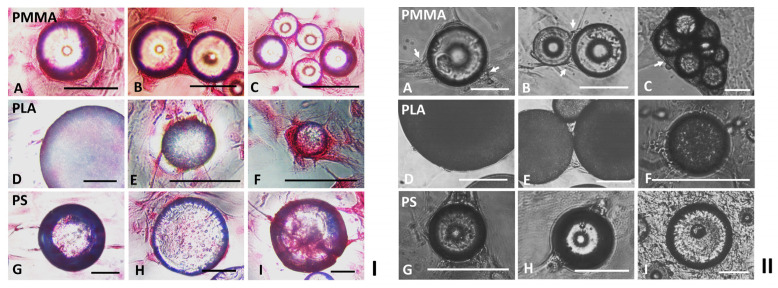
(**I**) Sirius red staining of HDFs cultured on the spherical modular scaffolds. HDFs cultured for 2–6 days in tissue culture plates with (**A**–**C**) PMMA, (**D**–**F**) PLA or (**G**–**I**) PS particles were fixed with 70% ethanol, stained with Sirius red, analyzed via phase contrast microscopy (PCM). (scale bar = 100 µm). (**II**) The influence of different polymeric particles on the cells cultured on the flat surfaces of 24-well culture plates. HDFs cultured for more than 6–10 days in the tissue culture plates with (**A**–**C**) PMMA, (**D**–**F**) PLA or (**G**–**I**) PS particles were analyzed via PCM. The cells peeled off by PMMA particles from the flat plate surfaces are highlighted via White arrows. (scale bar = 100 µm).

**Figure 3 ijms-24-05234-f003:**
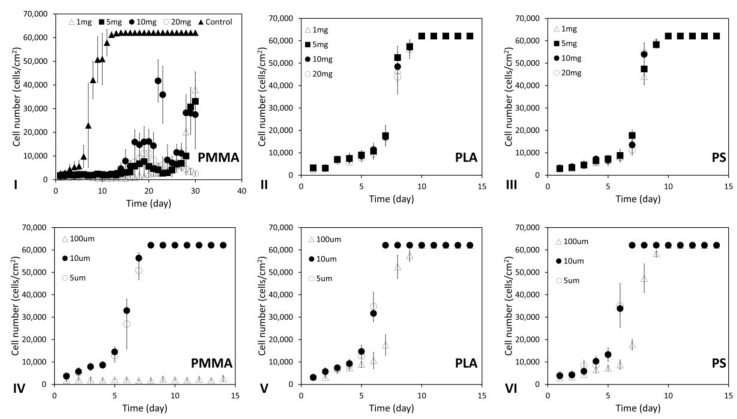
(**I**–**III**) The influences of particle material and number on the neighboring 2D cell cultures. Aliquots of 1 mL of HDFs (10^5^ cells/mL) were seeded into each of T25 flasks containing 1, 5, 10 or 20 mg of (**I**) PMMA, (**II**) PLA or (**III**) PS particles with defined diameter (100 µm) in 5 mL DMEM media and cultured for 14–30 days. (**IV**–**VI**) The influences of particle material and size on the neighboring 2D cell cultures. The same procedure was used to culture HDFs in each of T25 flasks containing 5 mg of (**IV**) PMMA, (**V**) PLA or (**VI**) PS particles with different diameters (5–100 µm) for 14 days. The cells on the flat surfaces of T25 flasks were estimated on daily basis via PCM. In all these experiments, T25 flasks containing no polymeric particles were used as the controls. Results shown are mean ± standard deviation (SD) (*n* = 3 independent experiments).

**Figure 4 ijms-24-05234-f004:**
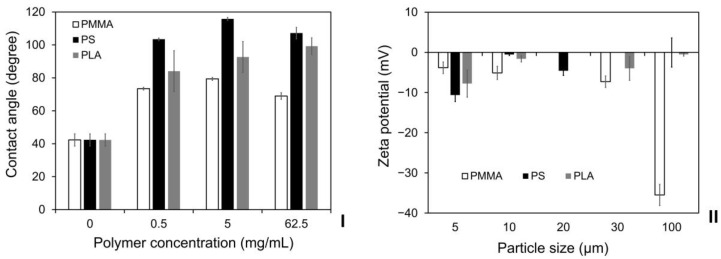
(**I**) The influences of polymeric materials and concentrations on particle wettability. (**II**) The influences of polymeric materials and particle diameter on zeta potential measurement in deionized water. White: PMMA; Black: PS; Grey: PLA. Results shown are mean ± SD (*n* = 3 independent experiments).

**Figure 5 ijms-24-05234-f005:**
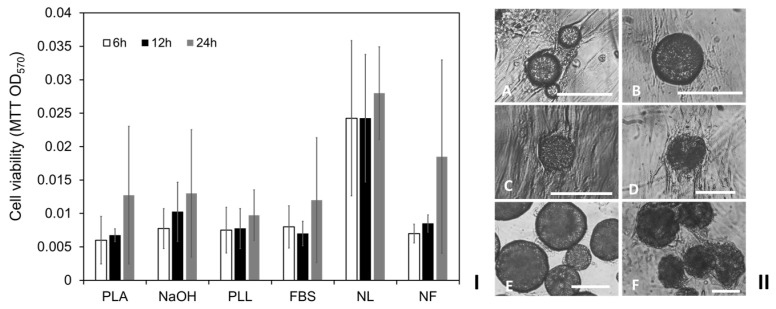
Influences of PLA particle surface modifications on modular tissue cultures. (**I**) MTT viability assay (OD_570_) of HDFs seeded and cultured for 6, 12 and 24 h on PLA particles modified using different methods: PLA: not-treated PLA particle control, NaOH: hydrolysis in 0.5 M sodium hydroxide (NaOH), PLL: coated with poly-L-lysine (PLL); FBS: coated with fetal bovine serum (FBS), NL: PLL coating after NaOH hydrolysis; NF: FBS coating after NaOH hydrolysis. Results shown are mean ± SD (*n* = 3 independent experiments). (**II**) Micrographs of the HDFs cultured in each well of 24-well tissue culture plates with PLA particles modified using different methods: (**A**) PLA, (**B**) FBS, (**C**) PLL, (**D**) NaOH, (**E**) NF, (**F**) NL. Scale bar = 100 µm.

**Figure 6 ijms-24-05234-f006:**
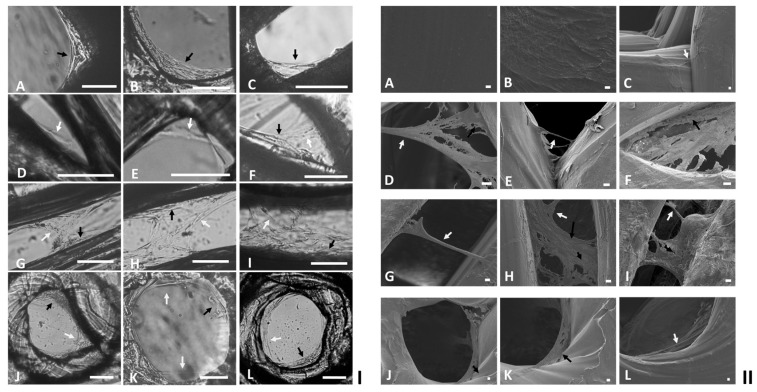
Micrographs of HDFs cultured on PLA discs with finely controlled corners, gaps or circular open pores. Aliquots of 50 µL HDFs (10^6^ cells/mL) were seeded on the top surfaces of PLA discs for 1 h for firm cell attachment, then cultured in 1 mL DMEM media for 40–60 days and analyzed via (**I**) PCM (scale bars = 100 µm), or (**II**) SEM (scale bars = 10 µm). Cells colonized the regular open structures via cell bridging (White arrows) or cell-stacking (Black arrows) strategies are highlighted.

**Table 1 ijms-24-05234-t001:** Parameters for the fabrication of polymeric particles.

Particle Diameter (µm)	100	30	10	5
Solvent	DCM ^1^	Chloroform	Chloroform	Chloroform
Organic solution (mg/mL)	62.5	5	5	5
PVA ^2^ (%(*w*/*v*))	5	0.1	1	5
O/A ^3^ ratio	8:150	1:10	1:10	1:10
Stirring speed (RPM)	200	1500	1500	1500
Temperature (°C)	RT ^4^	65	65	65
Evaporation time (h)	2.5	1.5	1.5	1.5

^1^ DCM: Dichloromethane; ^2^ PVA: Poly(vinyl alcohol); ^3^ O/A: Organic/aqueous; ^4^ RT: Room temperature.

**Table 2 ijms-24-05234-t002:** 3D Printing parameters for the PLA discs.

Printing Parameters	Set Values
Nozzle temperature (°C)	210
Print bed temperature (°C)	60
Printing speed (mm/min)	1000
Set extrusion width (mm)	0.4
Layer height (mm)	0.167

## Data Availability

Publicly available datasets were analyzed in this study.

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
