# Peer review of "Evaluation of Polymeric Particles for Modular Tissue Cultures in Developmental Engineering"

_ijms, 2023, doi:10.3390/ijms24065234_

Round 1

Reviewer 1 Report

This paper provides a finding that tissue morphology can depend on particle size and surface formulations such as charge and hydrophobicity. The overall research area is significant because cells cultured by such methods can reveal features mimicking a tissue, and so such methods may be used for tissue engineering purpose in a long run. In this paper, the authors develop some polymer-based spherical modulator scaffolds to examine cell and tissue morphology on such scaffolds. Specifically, they find that HDF cells cultured on submerged large-size PMMA particles tend to generate a 3D aggregate structure, which presumably enables cells to grow in a dynamic seeding bioreactor. The findings potentially reveal the significance of microcarriers for tissue engineering purpose. However, some important control data (such as no polymers) are missing, and the transition among sections and paragraphs are not properly addressed. The author should consider rewriting to clarify reasons of carrying out why they focus on such experiments rather than other alternatives.

1.    In Figure 1: In Panel I, the used colors for fluorescent micrographs are not labeled. Furthermore, the authors may consider putting polymer names with figures. In Panel II, it is unclear to me how the authors tell if the bridges are from single cells or multiple cells.

2.    In Figure 2: Do you think if the microspheres impact ECM remodeling? Furthermore, the authors may consider comparison of microcarrier-supported cells vs. 3D-cultured spheroids to improve biological insights. Do these cells generate polarity upon developing around microspheres?

3.    In Figure 3 and 4: The control data (without polymers) are missing, which makes the argument difficult to evaluate if such particles are contributing to those biological and physical readouts. In addition, the author may consider writing down polymer names in each panel.

4.    In Line 197-198: Do you have any explanation of why aggregate tendency is inversely proportional to hydrophobicity? Is it due to the nature of cell adhesion? Alternatively, I would expect the hydrophobic spheres would form aggregates easily due to entropic forces.

5.    It is unclear why the authors choose PLA for modifications but not PMMA that shows more dramatic effects in earlier figures.

Author Response

The following are our responses to the reviewers’ comments and corresponding revisions made in the manuscript:

Review #1:

General comments:

This paper provides a finding that tissue morphology can depend on particle size and surface formulations such as charge and hydrophobicity. The overall research area is significant because cells cultured by such methods can reveal features mimicking a tissue, and so such methods may be used for tissue engineering purpose in a long run. In this paper, the authors develop some polymer-based spherical modulator scaffolds to examine cell and tissue morphology on such scaffolds. Specifically, they find that HDF cells cultured on submerged large-size PMMA particles tend to generate a 3D aggregate structure, which presumably enables cells to grow in a dynamic seeding bioreactor. The findings potentially reveal the significance of microcarriers for tissue engineering purpose. However, some important control data (such as no polymers) are missing, and the transition among sections and paragraphs are not properly addressed. The author should consider rewriting to clarify reasons of carrying out why they focus on such experiments rather than other alternatives.

Responses to the general comments: We highly appreciate the reviewer’s positive comments on the significant implications of our research such as the mimicking of tissue structures via manipulating the sizes and surface properties of different polymeric particles, the potential to use current microcarrier-based technology for the manufacturing of modular tissues at large scale. Based on the constructive suggestions, we have modified the manuscript accordingly. The important control data, i.e. 2D cell cultures in T25 flasks without any polymeric particles have been included in Figure 3I, in the figure legend (lines 187-188). It has also been included and explained in different sections of the results, i.e. lines 160-165, lines 194-198.   

Justifications have been provided at different parts of the manuscript, i.e. lines 71-75, lines 191-192, lines 213-214, lines 220-221, lines 239-241, lines 271-273. The details are used to explain the transitions among different sections and paragraphs, and also used to clarify and justify the designed experiments rather than other alternatives.

The following are our responses to the specific comments:

Point 1: In Figure 1: In Panel I, the used colors for fluorescent micrographs are not labeled. Furthermore, the authors may consider putting polymer names with figures. In Panel II, it is unclear to me how the authors tell if the bridges are from single cells or multiple cells.

Response to Point 1: As suggested, the colors of fluorescent micrographs (Blue: nuclei stained by DAPI, Green: cytoskeleton stained by phalloidin) have been explained in the legend of Figure 1, in lines 120-121. The polymers in Figures 1I-II are also labelled. In Panel II, the single-cell bridges and multiple-cell bridges were initially estimated via comparing the size of the cell cultures in phase contrast micrographs, but they were mainly distinguished by the number of DAPI stained nuclei (Blue) in the fluorescent micrographs, i.e. in the single-cell bridges, only one DAPI stained nucleus was identified, while in the multiple-cell bridges more DAPI stained nuclei were identified. These cell structures were then checked via SEM with higher resolutions. For the ease of reading and understanding, it is also explained in lines 123-124.

Point 2: In Figure 2: Do you think if the microspheres impact ECM remodeling? Furthermore, the authors may consider comparison of microcarrier-supported cells vs. 3D-cultured spheroids to improve biological insights. Do these cells generate polarity upon developing around microspheres?

Response to Point 2: In this research, it was observed that HDFs could migrate from the flat surfaces of the tissue culture plates onto the PMMA and PLA particles, which was within our expectations. Interestingly, HDFs also migrated onto the PS particles, which were not ideal for cell cultures. Moreover, the cells on the PS particles also expressed collagens (as shown in Figure 2, panel I). Therefore, it can be deduced that HDFs could potentially modify the particle surfaces via synthesis of ECM proteins such as collagen, i.e. the investigated particles did demonstrate some impacts on ECM remodeling. As the reviewer kindly suggested, the comparison of microcarrier-supported cells vs. 3D-cultured spheroids will further improve the biological insights in cell-scaffold interactions and tissue regenerations. However, the focus of this research was to investigate the influences of particle materials and sizes on modular tissue cultures. Thus, PMMA, PLA and PS with distinct properties were selected. Moreover, the surface properties of these PMMA, PLA and PS particles are clearly defined and can be further purposefully modified. The particles with different sizes can be synthesized using the same fabrication method for systematic comparisons. In contrast, the sizes of most of the commercial microcarriers are usually fixed, the surfaces of some of the microcarriers are also modified or optimized for cell cultures. Thus, systematic comparisons of microcarrier-supported cells vs. 3D-cultured spheroids will require careful selection of the microcarriers and the consideration of more elements (such as the ECM proteins coated, and the surface topographies) associated with some of microcarriers, which should be conducted in separate research projects. Microscopic analysis in this research suggested that HDFs did generate obvious polarity when the cells initially migrated from the tissue culture surfaces onto the particles, and migrated around the particles (as clearly shown in the SEM images).

Point 3: In Figure 3 and 4: The control data (without polymers) are missing, which makes the argument difficult to evaluate if such particles are contributing to those biological and physical readouts. In addition, the author may consider writing down polymer names in each panel.

Response to Point 3: As kindly suggested, the control group data (2D cell cultures in T25-flasks without the presence of any polymeric particles) has been added in Figure 3I, which is also applicable for Figure 3II-VI. It has also been explained in the figure legend (lines 187-188). It has also been included and explained in different sections of the results, i.e. lines 160-165, lines 194-198. The control group (i.e. the glass slide without any casted polymers or 0 concentration of polymers) for contact angle measurements has been added in Figure 4I, it has also been explained and described in the corresponding result section (in lines 215-218), as well as described in section 4.11 of 4. Materials and Methods ( in lines 484-485). The polymer names have all been included in Figures 2-4 as suggested.

Point 4:  In Line 197-198: Do you have any explanation of why aggregate tendency is inversely proportional to hydrophobicity? Is it due to the nature of cell adhesion? Alternatively, I would expect the hydrophobic spheres would form aggregates easily due to entropic forces.

Response to Point 4: The aggregation tendency of polymeric particles is dependent on the initial entropic forces or hydrophobicity, the further cell adhesion and colonization. It also depends on the denaturation or conformational changes of the proteins adsorbed from the cell culture medium (e.g. the supplemented fetal bovine serum) onto the hydrophilic or hydrophobic particles, when the particles were initially submerged in the DMEM media as suggested in the added reference (i.e. [70]). This has been further explained in lines 358-362. Meanwhile, these absorbed proteins can further influence cell infiltration and colonization on these particles.

Point 5: It is unclear why the authors choose PLA for modifications but not PMMA that shows more dramatic effects in earlier figures.

Response to Point 5: As further explained in lines 239-241, due to its hydrophilicity which was detected to be lower than PMMA, but higher than PS, PLA was selected for further surface modifications and then compared with the non-treated particles. Surface modification of PLA particles will enable us to not only improve our understanding of particle aggregations, but also use these biodegradable PLA particles as the suitable modular scaffolds for manufacture of modular tissues.

Reviewer 2 Report

The manuscript is devoted to the study of  PMMA, PLA and PS particles with different sizes as scaffolds on the tissue cultures growth. It's a good job with a variety of methods involved, however there are some comments and questions before publication.

1. There are a lot of superfluous words in the Abstract, the conclusions and the essence of the work are not clearly presented.

2. There is no explanation  for the choice of polymers. Natural polymers are often used as scaffolds: polypeptides, polysaccharides, alginate, fibroin, polyesters, and so on. It is known that hydrofilization of scaffolds often promotes cell growth and proliferation. Why hydrophobic polymers PLA and PS were chosen? It seems quite obvious that hydrophobic polymer particles aggregate in a hydrophilic medium.

3. In point 4. Materials and Methods " poly(vinyl acetate) (PVA, Mw: 1300023000 g/mol, 87-89% hydrolyzed)"  is mentioned. Does hydrolyzed PVA mean polyvinyl alcohol  -  [CH2CH(OH)]n?

4. The surface topography (relief) is known to influence cell growth and proliferation. Perhaps, for complete conclusions, the authors should study the relief of particles by atomic force microscopy?

Author Response

The following are our responses to the reviewers’ comments and corresponding revisions made in the manuscript:

Review #2:

General comments:

The manuscript is devoted to the study of PMMA, PLA and PS particles with different sizes as scaffolds on the tissue cultures growth. It's a good job with a variety of methods involved, however there are some comments and questions before publication.

Response to the general comments: We highly appreciate the constructive comments on our research from the reviewer, particularly the use of various research methods. As suggested, we have made multiple modifications to further improve the quality of this manuscript.

The following are our responses to the specific comments:

Point 1: There are a lot of superfluous words in the Abstract, the conclusions and the essence of the work are not clearly presented.

Response to Point 1: We kindly acknowledge the comments. As no specific sentences or superfluous words are pointed out, we have made multiple modifications in the abstract, the conclusions and the essence of the work to improve the quality of these sections.

Point 2: There is no explanation for the choice of polymers. Natural polymers are often used as scaffolds: polypeptides, polysaccharides, alginate, fibroin, polyesters, and so on. It is known that hydrofilization of scaffolds often promotes cell growth and proliferation. Why hydrophobic polymers PLA and PS were chosen? It seems quite obvious that hydrophobic polymer particles aggregate in a hydrophilic medium.

Response to point 2: We agree with the reviewer that natural polymers such as polypeptides, polysaccharides, alginate, fibroin, polyesters are also suitable and commonly used as the scaffold materials for tissue engineering. However, this research aimed to investigate the materials and size of polymeric particles on modular tissue cultures, hence three district materials were selected for the ease of systematic comparisons. Amorphous hydrophilic polymer (PMMA), hydrophobic polymer (PLA) with certain crystallinity, and amorphous hydrophobic polymers (PS) were selected for this research because they have distinct but very defined properties. Moreover, PMMA, PLA and PS particles with different sizes or diameters could be produced using the particle fabrication method established in our lab. Based on this research using synthetic polymers, natural polymers can be used to further investigate and optimize the suitable modular scaffold for modular tissue cultures and tissue assemblies. The explanation of why these three polymers were selected for this research has been added in lines 71-75. As we explained in our responses to the first reviewer’s comments, the aggregation of polymeric particles in a hydrophilic medium is dependent on multiple factors such as the initial entropic forces or hydrophobicity, the subsequent cell adhesion and colonization. It also depends on the denaturation or conformational changes of the proteins adsorbed from the cell culture medium (e.g. the supplemented fetal bovine serum) onto the hydrophilic or hydrophobic particles, when the particles were initially submerged in the DMEM media as suggested in the added reference (i.e. [70]). This has been further explained in lines 358-362. Meanwhile, these absorbed proteins can further influence the cell infiltration and colonization on these particles, and the particle aggregations.

Point 3: In point 4. Materials and Methods "poly(vinyl acetate) (PVA, Mw: 13000–23000 g/mol, 87-89% hydrolyzed)" is mentioned. Does hydrolyzed PVA mean polyvinyl alcohol - [CH2CH(OH)]n?

Response to point 3: The PVA mentioned in 4.1 of 4. Materials and Methods should be poly(vinyl alcohol), it has now been corrected in lines 378 and 393.

Point 4: The surface topography (relief) is known to influence cell growth and proliferation. Perhaps, for complete conclusions, the authors should study the relief of particles by atomic force microscopy?

Response to Point 4: We agree with the reviewer’s comment that surface topography is another element that could influence cell growths and proliferations, which has been investigated by other researchers including one of our co-authors (as shown in the added reference [70]). Since the focus of this research was to investigate the influences of particle materials and sizes on modular tissue cultures in DE, particles with smooth surfaces were deliberately fabricated to avoid the impacts of surface topographies. Since it is not the focus of this research, this factor has not been discussed in the manuscript.

Round 2

Reviewer 1 Report

Overall the clarity has been substantially improved in this revision, and so the paper is ready to be published.